# On Two Intuitionistic Fuzzy Modal Topological Structures

**Krassimir Atanassov [1,2]**, **Nora Angelova [3]**, **Tania Pencheva [4,*]**

1 Department of Bioinformatics and Mathematical Modelling, Institute of Biophysics and Biomedical Engineering, Bulgarian Academy of Sciences, Acad. Georgi Bonchev Str., Bl. 105, 1113 Sofia, Bulgaria
2 Intelligent Systems Laboratory, Prof. Dr. Assen Zlatarov University, 1 "Prof. Yakimov" Blvd., 8010 Burgas, Bulgaria
3 Faculty of Mathematics and Informatics, Sofia University, 5 James Bouchier Blvd., 1164 Sofia, Bulgaria
4 Department of QSAR and Molecular Modelling, Institute of Biophysics and Biomedical Engineering, Bulgarian Academy of Sciences, Acad. Georgi Bonchev Str., Bl. 105, 1113 Sofia, Bulgaria
* Correspondence: tania.pencheva@biomed.bas.bg

**Abstract:** The concept of an Intuitionistic Fuzzy Modal Topological Structure (IFMTS) was introduced previously, and some of its properties were studied. So far, there are two different IFMTSs based on the classical intuitionistic fuzzy operations: "union" ($\cup$) and "intersection" ($\cap$). In the present paper, two new IFMTSs are developed. They are based on new intuitionistic fuzzy topological operators from closure and interior types, introduced here for the first time, and on the two standard intuitionistic fuzzy modal operators $\square$ and $\lozenge$. Some basic properties of the new IFMTSs are discussed. The newly presented IFMTSs could be considered as a basis for the next research on the IFMTSs. Some ideas for the future development of the IFMTS theory and open problems are formulated, related to the existence of other intuitionistic fuzzy operations that can generate new intuitionistic fuzzy topological operators and, respectively, new IFMTSs.

**Keywords:** Intuitionistic Fuzzy Modal Topology; Intuitionistic Fuzzy Operator; Intuitionistic Fuzzy Set

## 1. Introduction

The present paper is a continuation of the research in [1], in which the combination of the ideas and definitions from the areas of (general) topology (see, e.g., [2–4]), of (standard) modal logic (see, e.g., [5–8]) and of intuitionistic fuzziness (see, e.g., [9,10]) is introduced for first time. Presented there are two examples of Intuitionistic Fuzzy Modal Topological Structures (IFMTSs), based on the classical intuitionistic fuzzy operations "union" ($\cup$) and "intersection" ($\cap$). The reason for the present research is the authors' desire for new examples of such structures to be constructed that are different from those presented in [1]. The structures constructed in this investigation are based on the definitions of two new intuitionistic fuzzy topological operators from closure and interior types (in the terms of [3]). To date, and besides [1], the authors have recognized only the operations and topological operators presented in this paper as being able to satisfy all the conditions for IFTMS construction. In this line of reasoning, looking for new operators is still an open problem, as formulated in the Conclusion as well.

In Section 2, short remarks over Intuitionistic Fuzzy Sets (IFSs) are given, Section 3 presents the definitions of the new operators, while in Section 4, the new structures are introduced and some of their basic properties are discussed. In the Conclusion, some new directions of the development of the present ideas are discussed.

As mentioned in [1], the Intuitionistic Fuzzy Topology (IFT) has been developed very actively during the last years. The first steps in this process were published in [11–37], but until now there has been no systematic research on IFT development.

## 2. Short Remarks over IFSs

In 1983, the fuzzy sets of Lotfi Zadeh (1921–2017) [38] were extended to Intuitionistic Fuzzy Sets (IFSs, see, e.g., [9,10,39]). All results that are valid for the fuzzy sets can be transformed here as well.

Let a set $E$ be fixed. An IFS $A$ in $E$ is an object of the following form:

$$A = \{\langle x, \mu_A(x), \nu_A(x)\rangle | x \in E\},$$

where functions $\mu_A : E \to [0,1]$ and $\nu_A : E \to [0,1]$ define the degree of membership and the degree of non-membership of the element $x \in E$, respectively, and for every $x \in E$,

$$0 \le \mu_A(x) + \nu_A(x) \le 1.$$

Let the universe $E$ be given everywhere below. One of the geometrical interpretations of the IFSs is shown in Figure 1 (see [9,10]).

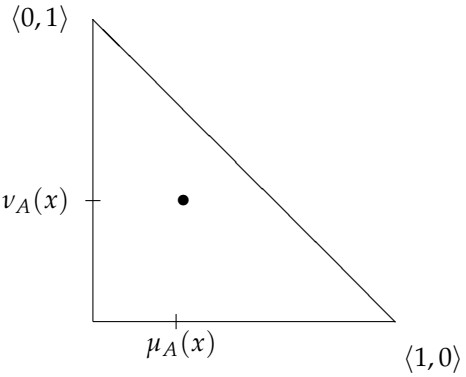

**Figure 1.** A geometrical interpretation of an element $x \in E$.

Following [10,40], it is important to mention that the functions $\mu, \nu$ (and also $\pi$, known as the degree of uncertainty and equal to $1 - \mu - \nu$) can be continuous or discrete with respect to the concrete cases. If the universe $E$ and the three functions are constructive objects, then the operations over the IFSs with the universe $E$ preserve the constructiveness.

For every two IFSs $A$ and $B$, a number of relations and operations are defined (see, e.g., [9,10]). The most important six of them are as follows:

$$A \subset B \quad \text{iff} \quad (\forall x \in E)(\mu_A(x) \le \mu_B(x) \,\&\, \nu_A(x) \ge \nu_B(x));$$

$$A \supset B \quad \text{iff} \quad B \subset A;$$
$$A = B \quad \text{iff} \quad (\forall x \in E)(\mu_A(x) = \mu_B(x) \,\&\, \nu_A(x) = \nu_B(x));$$

$$\neg A \quad = \quad \{\langle x, \nu_A(x), \mu_A(x)\rangle | x \in E\};$$

$$A \cap B \quad = \quad A \cap_4 B = \{\langle x, \min(\mu_A(x), \mu_B(x)), \max(\nu_A(x), \nu_B(x))\rangle | x \in E\};$$

$$A \cup B \quad = \quad A \cup_4 B = \{\langle x, \max(\mu_A(x), \mu_B(x)), \min(\nu_A(x), \nu_B(x))\rangle | x \in E\},$$

while the next two operations are introduced in [41]:

$$A \cap_{33} B \quad = \quad \{\langle x, \min(\mu_A(x), \mu_B(x)), 1 - \min(\mu_A(x), \mu_B(x))\rangle | x \in E\};$$

$$A \cup_{33} B \quad = \quad \{\langle x, 1 - \min(\nu_A(x), \nu_B(x)), \min(\nu_A(x), \nu_B(x))\rangle | x \in E\}.$$

The used subscripts 4 and 33 are introduced in [10,41] and express the subsequent number of the respective operation.

The last two operations do not have a direct analogue with those from the fuzzy set theory. The geometrical interpretations of these new operations are given for the first time in Figures 2 and 3.

In Section 3, the aforementioned last two operations will become a basis for the definitions of two new intuitionistic fuzzy topological operators.

The first two (simplest) analogues of the topological operators *closure* and *interior*, defined over IFSs, are presented as follows (see, e.g., [9,10]):

$$\mathcal{C}(A) = \{\langle x, \sup_{y \in E} \mu_A(y), \inf_{y \in E} \nu_A(y)\rangle | x \in E\},$$

$$\mathcal{I}(A) = \{\langle x, \inf_{y \in E} \mu_A(y), \sup_{y \in E} \nu_A(y)\rangle | x \in E\}.$$

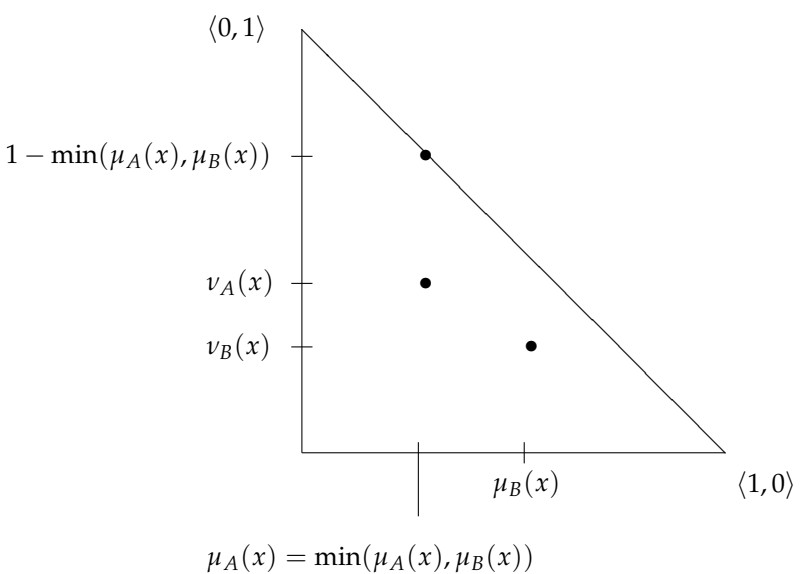

**Figure 2.** A geometrical interpretation of the operation $\cap_{33}$.

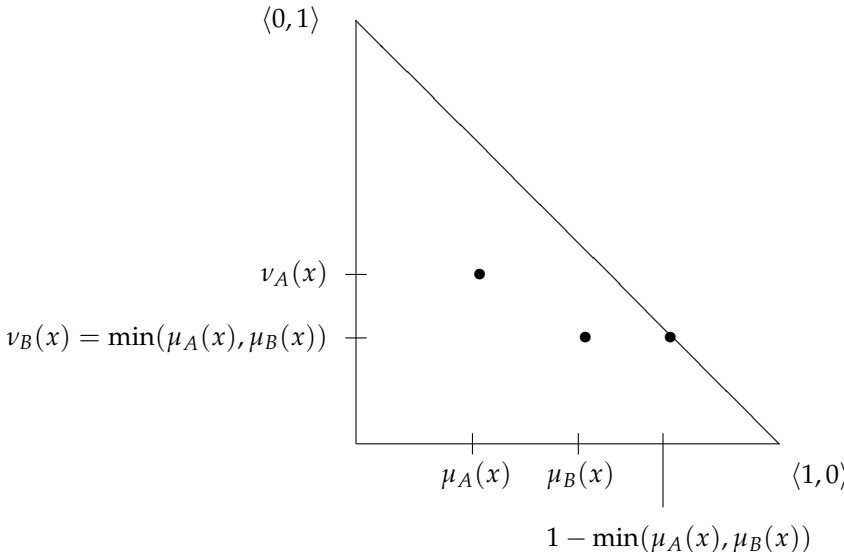

**Figure 3.** A geometrical interpretation of the operation $\cup_{33}$.

The geometrical interpretations of both intuitionistic fuzzy topological operators $\mathcal{C}(A)$ and $\mathcal{I}(A)$ are given in Figures 4 and 5.

Over the IFSs, different modal operators can be defined. They do not have analogues in fuzzy set theory.

The first two (simplest) modal operators, defined over the IFSs, are as follows (see, e.g., [5–8]):

$$\Box A = \{\langle x, \mu_A(x), 1 - \mu_A(x)\rangle | x \in E\};$$

$$\Diamond A = \{\langle x, 1 - \nu_A(x), \nu_A(x)\rangle | x \in E\}.$$

The geometrical interpretation of both intuitionistic fuzzy modal operators $\Box$ and $\Diamond$ is given in Figure 6.

Let

$$O^* = \{\langle x, 0, 1\rangle | x \in E\},$$

$$E^* = \{\langle x, 1, 0\rangle | x \in E\}.$$

Let for each set $X$

$$\mathcal{P}(X) = \{Y | Y \subseteq X\}.$$

Then, for sets $O^*$ and $E^*$, we obtain that

$$\mathcal{P}(O^*) = \{O^*\},$$

$$\mathcal{P}(E^*) = \{A | A \subseteq E^*\}.$$

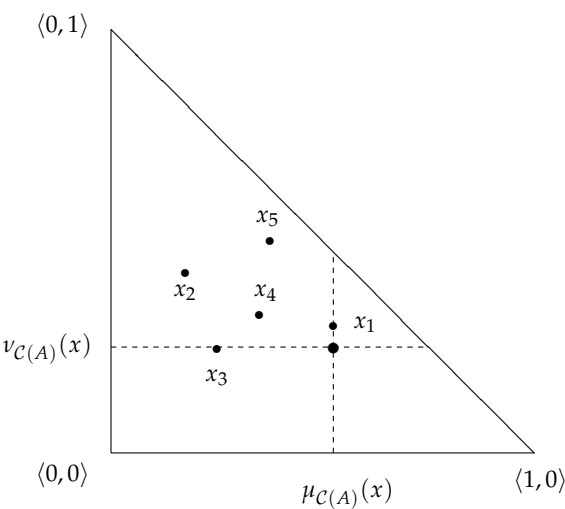

**Figure 4.** Geometrical interpretation of the topological operator $\mathcal{C}$.

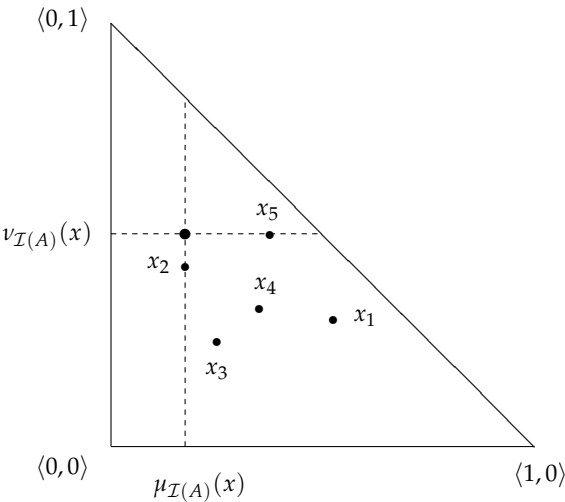

**Figure 5.** Geometrical interpretation of the topological operator $\mathcal{I}$.

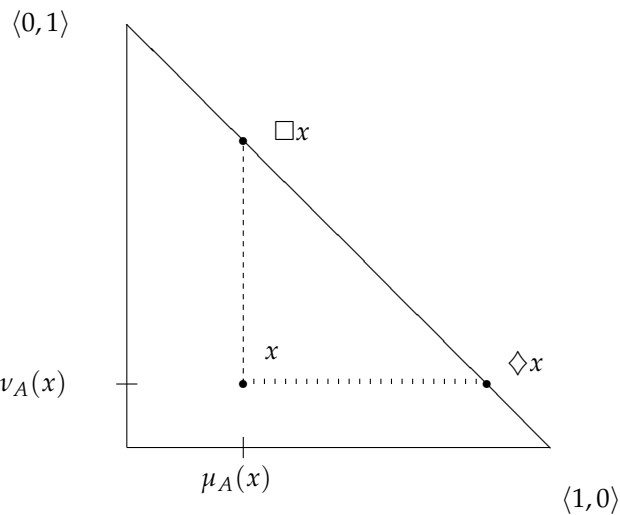

**Figure 6.** Geometrical interpretation of the modal operators $\square$ and $\diamondsuit$.

### 3. Definitions of Two New Intuitionistic Fuzzy Topological Operators

Using the method for the construction of the first two intuitionistic fuzzy topological operators $\mathcal{C}$ and $\mathcal{I}$ on the basis of the standard operations $\cup$ and $\cap$ (see [9,10]), below, we construct two new intuitionistic fuzzy topological operators. At the moment, the first two topological operators have no indices, but for completeness, we can index them as $\mathcal{C}_4$ and $\mathcal{I}_4$. The next operators are indexed using the number of intuitionistic fuzzy operations that generate them. Thus, the operators newly presented here are as follows for each IFS $A$:

$$\mathcal{C}_{33}(A) = \{\langle x, 1 - \inf_{y \in E} \nu_A(y), \inf_{y \in E} \nu_A(y)\rangle | x \in E\};$$

$$\mathcal{I}_{33}(A) = \{\langle x, \inf_{y \in E} \mu_A(y), 1 - \inf_{y \in E} \mu_A(y)\rangle | x \in E\}.$$

The geometrical interpretations of the two intuitionistic fuzzy topological operators are given in Figures 7 and 8.

**Theorem 1.** *For each IFS $A$:*

*(a)* $\neg\mathcal{C}_{33}(\neg A) = \mathcal{I}_{33}(A)$,

*(b)* $\neg\mathcal{I}_{33}(\neg A) = \mathcal{C}_{33}(A)$.

**Proof.** For case (a), we obtain

$$\begin{aligned}
\neg\mathcal{C}_{33}(\neg A) &= \neg\mathcal{C}_{33}(\{\langle x, \nu_A(y), \mu_A(y)\rangle | x \in E\}); \\
&= \neg\{\langle x, 1 - \inf_{y \in E} \mu_A(y), \inf_{y \in E} \mu_A(y)\rangle | x \in E\}; \\
&= \{\langle x, \inf_{y \in E} \mu_A(y), 1 - \inf_{y \in E} \mu_A(y)\rangle | x \in E\}; \\
&= \mathcal{I}_{33}(A).
\end{aligned}$$

The check of (b) is similar. $\square$

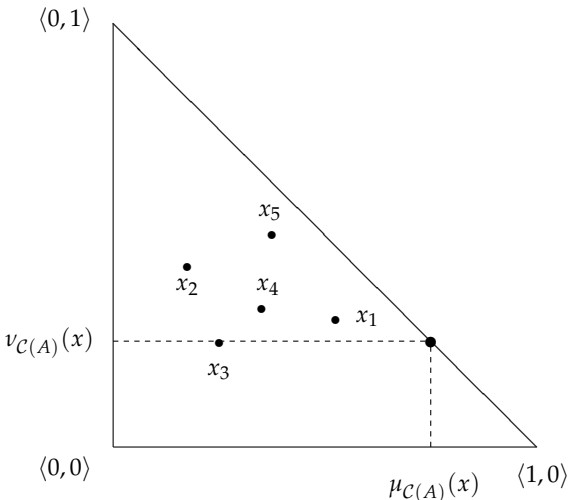

**Figure 7.** Geometrical interpretation of the topological operator $\mathcal{C}_{33}$.

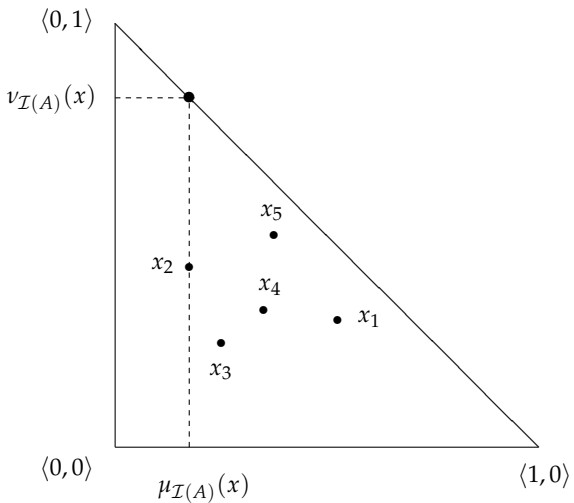

**Figure 8.** Geometrical interpretation of the topological operator $\mathcal{I}_{33}$.

**Theorem 2.** *For each IFS A:*

$$\mathcal{I}_{33}(A) \subseteq \mathcal{I}(A) \subseteq \mathcal{C}(A) \subseteq \mathcal{C}_{33}(A).$$

**Proof.** From the definitions of operators $\mathcal{C}$ and $\mathcal{I}$, for each $x \in E$, we have the equalities (see, e.g., [9,10])

$$0 \leq \sup_{y \in E} \mu_A(y) + \inf_{y \in E} \nu_A(y) \leq 1,$$

$$0 \leq \inf_{y \in E} \mu_A(y) + \sup_{y \in E} \nu_A(y) \leq 1.$$

Therefore,

$$\inf_{y \in E} \mu_A(y) \leq \sup_{y \in E} \mu_A(y) \leq 1 - \inf_{y \in E} \nu_A(y)$$

and

$$1 - \inf_{y \in E} \mu_A(y) \geq \sup_{y \in E} \nu_A(y) \geq \inf_{y \in E} \mu_A(y)$$

from which the validity of the assertion follows. $\square$

## 4. Definitions of the Two New Intuitionistic Fuzzy Modal Topological Structures and Their Examples

By analogy with the definition of Kazimierz Kuratowski (1896–1980) [3], in [1], we gave the definition of the IFMTS, which is an essential extension of the one in [3]. Here, we repeat the definition given in [1], but with a small correction as follows. The object

$$\langle \mathcal{P}(E^*), \mathcal{O}, \Delta, \circ \rangle,$$

is called a *cl*-IFMTS when the following are true:

- $E$ is a fixed universe,
- $\Delta : \mathcal{P}(E^*) \times \mathcal{P}(E^*) \to \mathcal{P}(E^*)$ is an operation over $E$,
- $\mathcal{O} : \mathcal{P}(E^*) \to \mathcal{P}(E^*)$ is an operator over $E$, generated by an operation $\Delta$,
- $\circ$ is a modal operator,

and for every two IFSs $A, B \in \mathcal{P}(E^*)$, the following conditions are valid:

C1  $\mathcal{O}(A\Delta B) = \mathcal{O}(A)\Delta\mathcal{O}(B)$,

C2  $A \subseteq \mathcal{O}(A)$,

C3  $\mathcal{O}(O^*) = O^*$,

C4  $\mathcal{O}(\mathcal{O}(A)) = \mathcal{O}(A)$,

C5  $\circ(A\nabla B) = \circ A\nabla \circ B$,

C6  $\circ A \subseteq A$,

C7  $\circ E^* = E^*$,

C8  $\circ \circ A = \circ A$,

C9  $\circ \mathcal{O}(A) = \mathcal{O}(\circ A)$.

In the definition in [1], an operation $\nabla$ was added in order to be dual according to the De Morgan Law with operation $\Delta$, and for this reason, it is omitted in the present definitions.

Having in mind the results from Section 3, we formulate and prove the following theorems.

**Theorem 3.** $\langle \mathcal{P}(E^*), \mathcal{C}_{33}, \cup_{33}, \diamondsuit \rangle$ *is a cl-IFMTS.*

**Proof.** Let the IFSs $A, B \in \mathcal{P}(E^*)$ be given. Then, we check sequentially the validity of the nine conditions C1–C9.

C1.

$$
\begin{aligned}
\mathcal{C}_{33}(A \cup_{33} B) &= \mathcal{C}_{33}(\{\langle x, \mu_A(x), \nu_A(x)\rangle | x \in E\} \cup_{33} \{\langle x, \mu_B(x), \nu_B(x)\rangle | x \in E\}) \\
&= \mathcal{C}_{33}(\{\langle x, 1 - \min(\nu_A(x), \nu_B(x)), \min(\nu_A(x), \nu_B(x))\rangle | x \in E\}) \\
&= \{\langle x, 1 - \inf_{y \in E}(\min(\nu_A(x), \nu_B(x))), \inf_{y \in E}\min(\nu_A(x), \nu_B(x))\rangle | x \in E\} \\
&= \{\langle x, 1 - \min(\inf_{y \in E}\nu_A(x), \inf_{y \in E}\nu_B(x)), \min(\inf_{y \in E}\nu_A(x), \inf_{y \in E}\nu_B(x))\rangle | x \in E\} \\
&= \{\langle x, 1 - \inf_{y \in E}\nu_A(x), \inf_{y \in E}\nu_A(x)\rangle | x \in E\} \\
&\quad \cup_{33}\{\langle x, 1 - \inf_{y \in E}\nu_B(x), \inf_{y \in E}\nu_B(x)\rangle | x \in E\} \\
&= \mathcal{C}_{33}(A) \cup_{33} \mathcal{C}_{33}(B);
\end{aligned}
$$

C2.

$$\begin{aligned}
A \ &= \{\langle x, \mu_A(x), \nu_A(x)\rangle | x \in E\} \\
&\subseteq \{\langle x, \sup_{y \in E} \mu_A(x), \inf_{y \in E} \nu_A(x)\rangle | x \in E\} \\
&\subseteq \{\langle x, \sup_{y \in E}(1 - \nu_A(x)), \inf_{y \in E} \nu_A(x)\rangle | x \in E\} \\
&= \{\langle x, 1 - \inf_{y \in E} \nu_A(x), \inf_{y \in E} \nu_A(x)\rangle | x \in E\} \\
&= \mathcal{C}_{33}(A);
\end{aligned}$$

C3.

$$\begin{aligned}
\mathcal{C}_{33}(O^*) \ &= \mathcal{C}_{33}(\{\langle x, 0, 1\rangle | x \in E\}) \\
&= \{\langle x, 1 - \inf_{y \in E} 1, \inf_{y \in E} 1\rangle | x \in E\} \\
&= \{\langle x, 0, 1\rangle | x \in E\} \\
&= O^*;
\end{aligned}$$

C4.  Having in mind that $\inf_{y \in E} \nu_A(y)$ is a constant, we obtain that

$$\begin{aligned}
\mathcal{C}_{33}(\mathcal{C}_{33}(A)) \ &= \mathcal{C}_{33}(\{\langle x, 1 - \inf_{y \in E} \nu_A(x), \inf_{y \in E} \nu_A(x)\rangle | x \in E\}) \\
&= \{\langle x, 1 - \inf_{z \in E} \inf_{y \in E} \nu_A(y), \inf_{z \in E} \inf_{y \in E} \nu_A(y)\rangle | x \in E\} \\
&= \{\langle x, 1 - \inf_{y \in E} \nu_A(y), \inf_{y \in E} \nu_A(y)\rangle | x \in E\} \\
&= \mathcal{C}_{33}(A);
\end{aligned}$$

C5.

$$\begin{aligned}
\Diamond(A \cup_{33} B) \ &= \Diamond(\{\langle x, 1 - \min(\nu_A(x), \nu_B(x)), \min(\nu_A(x), \nu_B(x))\rangle | x \in E\}); \\
&= \{\langle x, 1 - \min(\nu_A(x), \nu_B(x)), \min(\nu_A(x), \nu_B(x))\rangle | x \in E\}; \\
&= \{\langle x, \mu_A(x), \nu_A(x)\rangle | x \in E\} \cup_{33} \{\langle x, \mu_B(x), \nu_B(x)\rangle | x \in E\} \\
&= \Diamond A \cup_{33} \Diamond B;
\end{aligned}$$

C6.

$$\begin{aligned}
A \ &= \{\langle x, \mu_A(x), \nu_A(x)\rangle | x \in E\} \\
&\subseteq \{\langle x, \mu_A(x), 1 - \nu_A(x)\rangle | x \in E\} \\
&= \Diamond\{\langle x, \mu_A(x), \nu_A(x)\rangle | x \in E\} \\
&= \Diamond A;
\end{aligned}$$

C7.

$$\begin{aligned}
\Diamond E^* \ &= \Diamond\{\langle x, 1, 0\rangle | x \in E\}) \\
&= \{\langle x, 1 - 0, 0\rangle | x \in E\} \\
&= (\{\langle x, 1, 0\rangle | x \in E\}) \\
&= E^*;
\end{aligned}$$

C8.

$$\diamondsuit\diamondsuit A \quad = \diamondsuit\{\langle x, 1 - \nu_A(x), \nu_A(x)\rangle | x \in E\}$$
$$= \{\langle x, 1 - \nu_A(x), \nu_A(x)\rangle | x \in E\}$$
$$= \diamondsuit A;$$

C9.

$$\diamondsuit\mathcal{C}_{33}(A) \quad = \diamondsuit\{\langle x, \sup_{y \in E} \mu_A(y), \inf_{y \in E} \nu_A(y)\rangle | x \in E\})$$
$$= \{\langle x, \sup_{y \in E} \mu_A(y), 1 - \sup_{y \in E} \nu_A(y)\rangle | x \in E\})$$
$$= \mathcal{C}_{33}(\{\langle x, 1 - \nu_A(x), \nu_A(x)\rangle | x \in E\})$$
$$= \mathcal{C}_{33}(\diamondsuit A).$$

This completes the proof of Theorem 3. $\square$

By analogy with the definition of *cl*-IFMTS and of Kuratowski's definition for inclusion operator, here we state that the object

$$\langle \mathcal{P}(E^*), \mathcal{Q}, \nabla, \bullet \rangle,$$

is called a *in*-IFMTS when the following are true:

- *E* is a fixed universe,
- $\nabla : \mathcal{P}(E^*) \times \mathcal{P}(E^*) \to \mathcal{P}(E^*)$ is an operation over *E*,
- $\mathcal{Q} : \mathcal{P}(E^*) \to \mathcal{P}(E^*)$ is an operator over *E*, generated by operation $\nabla$,
- $\bullet$ is a modal operator,

and for every two IFSs $A, B \in \mathcal{P}(E^*)$, the following conditions are valid:

I1   $\mathcal{Q}(A\nabla B) = \mathcal{Q}(A)\nabla\mathcal{Q}(B)$,

I2   $\mathcal{Q}(A) \subseteq A$,

I3   $\mathcal{Q}(E^*) = E^*$,

I4   $\mathcal{Q}(\mathcal{Q}(A)) = \mathcal{Q}(A)$,

I5   $\bullet(A\nabla B) = \bullet A\nabla \bullet B$,

I6   $\bullet A \subseteq A$,

I7   $\bullet O^* = O^*$,

I8   $\bullet \bullet A = \bullet A$,

I9   $\bullet \mathcal{Q}(A) = \mathcal{Q}(\bullet A)$.

**Theorem 4.** $\langle \mathcal{P}(E^*), \mathcal{I}_{33}, \cap_{33}, \square \rangle$ *is a in-IFMTS.*

**Proof.** Let the IFSs $A, B \in \mathcal{P}(E^*)$ be given. Then, we check sequentially the validity of the nine conditions I1–I9.

I1.

$$\begin{aligned}
\mathcal{I}_{33}(A \cap_{33} B) &= \mathcal{I}_{33}(\{\langle x, \min(\mu_A(x), \mu_B(x)), 1 - \min(\mu_A(x), \mu_B(x))\rangle | x \in E\}) \\
&= \{\langle x, \inf_{y \in E}(\min(\mu_A(x), \mu_B(x))), 1 - \inf_{y \in E}(\min(\mu_A(x), \mu_B(x)))\rangle | x \in E\} \\
&= \{\langle x, \min(\inf_{y \in E}\mu_A(x), \inf_{y \in E}\mu_B(x)), 1 - \min(\inf_{y \in E}\mu_A(x), \inf_{y \in E}\mu_B(x))\rangle | x \in E\} \\
&= \{\langle x, \inf_{y \in E}\mu_A(x), 1 - \inf_{y \in E}\mu_A(x)\rangle | x \in E\} \\
&\quad \cap_{33}\{\langle x, \inf_{y \in E}\mu_B(x), 1 - \inf_{y \in E}\mu_B(x)\rangle | x \in E\} \\
&= \mathcal{I}_{33}(A) \cap_{33} \mathcal{I}_{33}(B);
\end{aligned}$$

I2.

$$\begin{aligned}
\mathcal{I}_{33}(A) &= \{\langle x, \inf_{y \in E}\mu_A(x), 1 - \inf_{y \in E}\mu_A(x)\rangle | x \in E\} \\
&= \{\langle x, \inf_{y \in E}\mu_A(x), \sup_{y \in E}(1 - \mu_A(x))\rangle | x \in E\} \\
&\subseteq \{\langle x, \inf_{y \in E}\mu_A(x), \sup_{y \in E}\nu_A(x)\rangle | x \in E\} \\
&\subseteq \{\langle x, \mu_A(x), \nu_A(x)\rangle | x \in E\} \\
&= A;
\end{aligned}$$

I3.

$$\begin{aligned}
\mathcal{I}_{33}(E^*) &= \mathcal{I}_{33}(\{\langle x, 1, 0\rangle | x \in E\}) \\
&= \{\langle x, \inf_{y \in E}1, 1 - \inf_{y \in E}1\rangle | x \in E\} \\
&= \{\langle x, 1, 0\rangle | x \in E\} \\
&= E^*;
\end{aligned}$$

I4.  Having in mind that $\sup_{y \in E}\nu_A(y)$ is a constant, we obtain that

$$\begin{aligned}
\mathcal{I}_{33}(\mathcal{I}_{33}(A)) &= \mathcal{I}_{33}(\{\langle x, \inf_{y \in E}\mu_A(x), 1 - \inf_{y \in E}\mu_A(x)\rangle | x \in E\}) \\
&= \{\langle x, \inf_{z \in E}\inf_{y \in E}\mu_A(x), 1 - \inf_{z \in E}\inf_{y \in E}\mu_A(x)\rangle | x \in E\} \\
&= \{\langle x, \inf_{y \in E}\mu_A(x), 1 - \inf_{y \in E}\mu_A(x)\rangle | x \in E\} \\
&= \mathcal{I}_{33}(A);
\end{aligned}$$

I5.

$$\begin{aligned}
\square(A \cap_{33} B) &= \square\{\langle x, \min(\mu_A(x), \mu_B(x)), 1 - \min(\mu_A(x), \mu_B(x))\rangle | x \in E\} \\
&= \{\langle x, \min(\mu_A(x), \mu_B(x)), 1 - \min(\mu_A(x), \mu_B(x))\rangle | x \in E\} \\
&= \{\langle x, \mu_A(x), 1 - \mu_A(x)\rangle | x \in E\} \cap_{33} \{\langle x, \mu_B(x), 1 - \mu_B(x)\rangle | x \in E\} \\
&= \square A \cap_{33} \square B;
\end{aligned}$$

I6.

$$\begin{aligned}
\square A &= \{\langle x, \mu_A(x), 1 - \mu_A(x)\rangle | x \in E\} \\
&\subseteq \{\langle x, \mu_A(x), \nu_A(x)\rangle | x \in E\} \\
&= A;
\end{aligned}$$

I7.

$$\begin{aligned}
\Box O^* &= \Box\{\langle x, 0, 1\rangle | x \in E\} \\
&= \{\langle x, 0, 1-0\rangle | x \in E\} \\
&= (\{\langle x, 0, 1\rangle | x \in E\}) \\
&= O^*;
\end{aligned}$$

I8.

$$\begin{aligned}
\Box\Box A &= \Box\{\langle x, \mu_A(x), 1-\mu_A(x)\rangle | x \in E\} \\
&= \{\langle x, \mu_A(x), 1-\mu_A(x)\rangle | x \in E\} \\
&= \Box A;
\end{aligned}$$

I9.

$$\begin{aligned}
\Box\mathcal{I}_{33}(A) &= \Box\{\langle x, \inf_{y\in E}\mu_A(x), 1-\inf_{y\in E}\mu_A(x)\rangle | x \in E\} \\
&= \Box\{\langle x, \inf_{y\in E}\mu_A(x), 1-\inf_{y\in E}\mu_A(x)\rangle | x \in E\} \\
&= \mathcal{I}_{33}(\{\langle x, \mu_A(x), \nu_A(x)\rangle | x \in E\}) \\
&= \mathcal{I}_{33}(\Box A).
\end{aligned}$$

This completes the proof of Theorem 4. $\square$

## 5. Conclusions or Ideas for the Future

In the present paper, two new IFMTSs are developed, which are based on the new Intuitionistic Fuzzy Topological Operators from closure and interior types and are different from the standard ones from [1] and recognized so far as the only ones able to satisfy the conditions of IFTMS constrictions. The basic properties of the new IFMTSs have been presented.

The ideas described in [1] and in the present investigation open some directions for the future development of the forms of the Intuitionistic Fuzzy Topological Structures (IFTSs). They can have not only the IFMTSs forms discussed in [1] and here but, e.g., those of Intuitionistic Fuzzy Temporal Topological Structures, introduced in [42].

An **open problem** is to search for new intuitionistic fuzzy conjunctions and disjunctions to generate new IFMTSs.

On the other hand, in some papers, e.g., [43], some of the conditions of the IFMTSs were changed with weaker ones (e.g., such that in which the relation "=" was changed with the relation "$\subseteq$" or "$\supseteq$"), or in which some of the conditions were omitted. These IFMTSs were called "feeble" IFMTSs. We use the word "feeble" to eliminate the conflict with the well-known concept of a weak topology. At the moment, six different types of such structures are introduced. So, an **Open Problem** is to search for other feeble IFMTSs.

Another **open problem** is what kinds of IFMTSs can be constructed on the basis of the extended modal operators (see [10]). In addition, in the IFSs theory, there are two other types of modal operators. Thus, another **open problem** is whether some of these modal operators can generate new IFMTSs.

Another direction of our research, mentioned in [1], is related to the extension of the forms of the topological operators discussed above. If, e.g., $\mathcal{A}(E) \subseteq \mathcal{P}(E^*)$, then we can construct the topological operators:

$$\overline{\mathcal{C}}(\mathcal{A}(E)) = \{\langle x, \sup_{A\in\mathcal{A}(E)}\mu_A(x), \inf_{A\in\mathcal{A}(E)}\nu_A(x)\rangle | x \in E\},$$

$$\overline{\mathcal{I}}(\mathcal{A}(E)) = \{\langle x, \inf_{A\in\mathcal{A}(E)}\mu_A(x), \sup_{A\in\mathcal{A}(E)}\nu_A(x)\rangle | x \in E\},$$

$$\overline{\mathcal{C}}_{33}(A(E)) = \{\langle x, 1-\inf_{A\in\mathcal{A}(E)}\nu_A(y), \inf_{A\in\mathcal{A}(E)}\nu_A(y)\rangle | A \in A(E)\},$$

$$\overline{\mathcal{I}}_{33}(A(E)) = \{\langle x, \inf_{A \in \mathcal{A}(E)} \mu_A(y), 1 - \inf_{A \in \mathcal{A}(E)} \mu_A(y)\rangle | A \in A(E)\}.$$

We can study the properties of these four new topological operators.

The next possible direction is related to the use of the extended topological operators, discussed in [10]. In practice, all of the above research can be re-written (in a more detailed form) for these extended topological operators, and we hope that new results, specific for them, will arise.

Finally, we must mention that in the IFSs theory, there are also level operators, which can be used as well for the construction of new IFTSs.

In the near future, the possibility to apply the above mentioned IFMTSs in different areas of data mining, intercriteria analysis and others will be investigated.

It is very worth noting that the **so defined conditions of IFMTS are conditions for each MTS, when the basic set is an arbitrary one, but not specifically an IFS**.

**Author Contributions:** Conceptualization, K.A. and N.A.; methodology, K.A. and N.A.; software, N.A.; validation, K.A., N.A. and T.P.; formal analysis, K.A., N.A. and T.P.; investigation, K.A., N.A. and T.P.; writing—original draft preparation, K.A., N.A. and T.P.; writing—review and editing, K.A., N.A. and T.P.; visualization, K.A. and N.A.; supervision, K.A.; project administration, K.A.; funding acquisition, K.A. All authors have read and agreed to the published version of the manuscript.

**Funding:** This research was funded by the Bulgarian National Science Fund grant number KP-06-N22-1/2018 "Theoretical Research and Applications of InterCriteria Analysis".

**Data Availability Statement:** Not applicable.

**Conflicts of Interest:** The authors declare no conflict of interest.

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
