# Peer review of "On Two Intuitionistic Fuzzy Modal Topological Structures"

_axioms, doi:10.3390/axioms12050408_

Round 1

Reviewer 1 Report

refer to the attached file review report of axioms-2283311.

Author Response

Thank you for your valuable comments and suggestions. Taking into account all of them, we believe that the quality of the paper has been improved significantly.

All improvements are presented in red color in order to be easily identifiable by the editors and reviewers.

Following are the detailed answers, point by point, to each of your comments.

To improve the manuscript, I give the following revisions and suggestions:

  1. The abstract should be improved carefully by comparing the new IFMTSs with the previous one.

We thank the Reviewer for this comment. The abstract has been improved as per Reviewer’s suggestions.

  1. The contribution is not clear. In the introduction, the authors should try to explain their research motivation, the original idea, and emphasize the unique contribution of this paper. Where is the research gap? Why do you propose two new IFMTSs? What is the shortcoming of the previous one? What is your contribution?

We thank the Reviewer for this comment. We changed significantly the beginning of the Introduction and thus we hope to meet the Reviewer’s expectations.

  1. Please define and list all the used symbols at the start of the manuscript.

We thank the Reviewer for this comment. All used symbols are defined and illustrated in a proper manner at the place of their first appearance in Section 2 (Short remarks over IFSs), while the definitions of two new operators age given in the Section 3. We do believe that this manner of presentation is more convenient to the readers than the listing of all symbols at the beginning of the manuscript before their detailed explanations.

4.The author can give an experiment or an example to illustrate the application of the proposed theory. This makes the article fuller and more logically persuasive.

We thank the Reviewer for this comment. The main theoretical result in this investigation is the idea and the definition of IFMTS. Indeed, Theorems 3 and 4 are the illustrations of the two newly constructed structures. In Conclusion we add the important remark that the so defined conditions of IFMTS are conditions for each MTS, when the basic set is an arbitrary one, but not specifically an intuitionistic fuzzy set.

5.The conclusion should be enhanced to summarize the main points of the manuscript. The limitations of this study should also be provided in this section.

We thank the Reviewer for this comment. The main contributions of our research have been added in the Conclusion. Concerning the limitation of this study – they will become clear when all intuitionistic fuzzy operations, which can be basis for the respective operators, be investigated. For the Reviewer information only, up to the moment we have about 600 different intuitionistic fuzzy unions and intersections, but so far we have not found no other to satisfy the conditions for IFTMS construction. In a series of about five published (or in press) papers different IFMTSs have been introduced, which did not fulfil all of the conditions, thus why they are called “feeble”.

  1. The operations proposed in [41] is a fuzzy set, is not a intuitionistic fuzzy set, since ….
  2. similar to …. and new definition in this paper….

We thank the Reviewer for these comments. The observation is quite exact, but as one can see, in some conditions (e.g., C2, C6, I2, I6) proper IFSs are existed. In case of fuzzy sets, strong “=” will exist in these conditions.

Reviewer 2 Report

This is a paper on intuitionistic fuzzy modal topological structures (abbreviated as IFMTS), written by a well-known mathematician in this domain K.Atanassov (Scopus H=28) and two younger co-authors (h=5 and 16). The research direction of the paper is related to some areas of general  topology, modal logic, and intuitionistic fuzziness. Two new examples of IFMTS, based on accordingly two new intuitionistic fuzzy topological operators, are defined and studied. The main results are theorems 1,2,3,4.

The introduction should be revised in order to better highlight the motivations and aims of the research. In particular, it is necessary to further highlight, with respect to the existing literature, what exactly problems motivated the research and what exactly are the innovative aspects introduced by the results demonstrated in the research. This is necessary to evaluate whether and why the paper generally warrants of publication by its content. The current introduction is too brief and non-informative, and thereby hardly fits to such a Journal as Axioms.

On the other hand, the conclusion (section 5) is way broader that the content of the paper. The authors should consider to narrower it and accordingly to get the open problems there more focused.

Line 28:
`(the second in time)'
not quite clear what does it mean

Between lines 35 and 36:
to have displayed lines numbered, the lineno package can be used

Between lines 35 and 36:
what is 4 in \cap_4 and \cup_4 ?
this is likely a concept from an earlier paper, so it would be useful to explain its meaning by a sentence or two.

Line 36
`are comparatively new ones'
This obscures things. The notions are either new (defined here) or are known from an earlier source.

Between lines 35 and 37
you define
A \cap_{33} B = .....min.....1-min
A \cup_{33} B = .....1-min.....min
First, it should be explained what is the meaning of 33.
Secondly, I am rather curious to know does it make sense to define
A *** B = .....1-min.....1-min

Lines 73-74-75
A short description of the small `o' (as in the last line on page 6) has to be given

Line 94
`This completes the proof'
add: of theorem 3 (?)

Page 10 in the middle
change I.6 to I6.

Page 10 item I7.
The first line ends by a right bracket ) which seems not to have a corresponding left (

Line 113
`This completes the proof'
add: of theorem 4 (?)

Line 118
`They can have not only the discussed in [1] and here forms as IFMTSs'
change to
They can have not only the IFMTS forms discussed in [1] and here
That is: first goes FORMS then goes DISCUSSED IN

To conclude, I believe that the paper could be published after rather minor corrections. Yet the text has to be carefully looked through once again, and likely rewritten here and there.

Author Response

Thank you for your valuable comments and suggestions. Taking into account all of them, we believe that the quality of the paper has been improved significantly.

All improvements are presented in red color in order to be easily identifiable by the editors and reviewers.

Following are the detailed answers, point by point, to each of your comments.

This is a paper on intuitionistic fuzzy modal topological structures (abbreviated as IFMTS), written by a well‐known mathematician in this domain K.Atanassov (Scopus H=28) and two younger co‐authors (h=5 and 16). The research direction of the paper is related to some areas of general topology, modal logic, and intuitionistic fuzziness. Two new examples of IFMTS, based on accordingly two new intuitionistic fuzzy topological operators, are defined and studied. The main results are theorems 1,2,3,4.

The introduction should be revised in order to better highlight the motivations and aims of the research. In particular, it is necessary to further highlight, with respect to the existing literature, what exactly problems motivated the research and what exactly are the innovative aspects introduced by the results demonstrated in the research. This is necessary to evaluate whether and why the paper generally warrants of publication by its content. The current introduction is too brief and non‐informative, and thereby hardly fits to such a Journal as Axioms.

We thank the Reviewer for this comment. We changed significantly the beginning of the Introduction and thus we hope to meet the Reviewer’s expectations. In fact, Section 2 contains preliminary information which, in many cases, may be considered as an introduction. But we considered its presentation as a separate section as more appropriate for better understanding.

On the other hand, the conclusion (section 5) is way broader that the content of the paper. The authors should consider to narrower it and accordingly to get the open problems there more focused.

We thank the Reviewer for this comment. Following its suggestion, two of the paragraphs have been significantly narrowed. Meanwhile, following the suggestion of another Reviewer, a new part has been introduced.

Line 28:

`(the second in time)'

not quite clear what does it mean

We thank the Reviewer for this comment. We omitted this explanation, but for the Reviewer information only, this is the second one chronologically (as for the moment, from more than 10) geometric interpretation.

Between lines 35 and 36:

to have displayed lines numbered, the lineno package can be used

We thank the Reviewer for this comment. The numbering comes with the journal template and will not appear in the case of paper acceptance. So we prefer do not interfere over the template.

Between lines 35 and 36:

what is 4 in \cap_4 and \cup_4 ?

this is likely a concept from an earlier paper, so it would be useful to explain its meaning by a sentence or two.

Line 36

`are comparatively new ones'

This obscures things. The notions are either new (defined here) or are known from an earlier source.

Between lines 35 and 37

you define

A \cap_{33} B = .....min.....1‐min

A \cup_{33} B = .....1‐min.....min

First, it should be explained what is the meaning of 33.

We thank the Reviewer for these comments. All the remarks so far have been taken into account and explained appropriately.

Secondly, I am rather curious to know does it make sense to define

A *** B = .....1‐min.....1‐min

We thank the Reviewer for this comment. In fact, if we have 1‐min (a, b), 1- min (c, d), then for
a = b = c = d = 0 we will obtain that the sum of both components will become 2, which is a contradiction to IFS definition. 

We thank the Reviewer for all the remarks below. All they have been corrected appropriately.

Lines 73‐74‐75

A short description of the small `o' (as in the last line on page 6) has to be given

Line 94

`This completes the proof'

add: of theorem 3 (?)

Page 10 in the middle

change I.6 to I6.

Page 10 item I7.

The first line ends by a right bracket ) which seems not to have a corresponding left (

Line 113

`This completes the proof'

add: of theorem 4 (?)

Line 118

`They can have not only the discussed in [1] and here forms as IFMTSs'

change to

They can have not only the IFMTS forms discussed in [1] and here

That is: first goes FORMS then goes DISCUSSED IN

To conclude, I believe that the paper could be published after rather minor corrections. Yet the text has to be carefully looked through once again, and likely rewritten here and there.

We thank the Reviewer for the overall high evaluation of our investigation.

Reviewer 3 Report

This original research article is within the scope of Axioms Journal.

The issue of intuitionistic fuzzy sets is very actual and challenging, since these structures can be used in various application domains.

The present state of knowledge and objectives have been very clearly presented.

In introduction, the authors describe short remarks over intuitionistic fuzzy sets.

The authors proposed two new intuitionistic fuzzy modal topological operators. The results of the study are correctly presented and interpreted. Moreover, they present the ideas for the future research.

The article is written clearly with a very good English. I appreciate the quality of 8 figures, which are very clearly presented. Up-to-date references are included, and the most relevant reports are cited.

Congratulation to a well-designed and performed study.

This original and novel research paper is valuable from the scientific point of view and I highly recommend accepting the paper without revision.

Author Response

We thank the Reviewer for the overall high evaluation of our investigation.

Round 2

Reviewer 1 Report

refer to attached file: review report of axioms-2283311-r1, the author cannot give any revision for their views.

Author Response

We thank the Reviewer for this comment, which was already posted on the first round of revisions.

In the revised version, sent after first round of revisions, in the Introduction we were added an explanation that “Up to the moment and besides [1], the authors have recognized only the operations and topological operators presented in this paper as able to satisfy all the conditions for IFTMS construction.” We hoped that with such explanation we answered the Reviewer’s comment. However, for the information to the Reviewer only, we would like to mention again, that the main contribution of this paper is that this is chronologically the second, but the only one structure found so far, different from the one presented in [1], which satisfies all the conditions for IFMTS construction. In fact, these operations generate not only IFS, but also a standard FS. After the detailed check we have not found any other pair “conjunction/disjunction”, that can generate topological operators, that to generate IFMTS. As such, this is formulated as an Open problem in the Conclusion.

Reviewer 2 Report

The authors have revised the text accordingly to my remarks. Now I believe it can be published in "Mathematics".
The following is not assumed to be really crusial so there is no need to return it to me once again.

line 17: here the abbreviation IFMTS occurs 1st time (not counting the abstract), so I believe it has to also be given in full.

Author Response

We thank the Reviewer for his evaluation and for his comment. The corresponding change has been done, marked in green in the revised version after second round of reviews.

Round 3

Reviewer 1 Report

it could be accepted for publish.